# Analysis of Cultural Meme Characteristics for Big Data of Cultural Relics

**Haifeng Li** [1] **, Zuoqin Shi** [1] **, Li Chen** [1,*] **, Zhenqi Cui** [1] **, Sumin Li** [2] **and Ling Zhao** [1]

[1] School of Geosciences and Info-Physics, Central South University, Changsha 410083, China; lihaifeng@csu.edu.cn (H.L.); szq1024@csu.edu.cn (Z.S.); cuizhenqi@csu.edu.cn (Z.C.); zhaoling@csu.edu.cn (L.Z.)

[2] School of Architecture, Changsha University of Science and Technology, Changsha 410000, China; lisumin57@gmail.com

[*] Correspondence: vchenli@csu.edu.cn

**Abstract:** The cultural meme is the smallest unit constituting a dynasty's culture, which has the same inheritance and variability as biological genes. Here, based on the name of cultural relics, we extract cultural memes through semantic word segmentation, word frequency statistics, and the synonym merging method, and construct dynasty cultural meme vectors. We analyzed color, auxiliary, texture, shape, and overall networks of five types of model to construct the culture network, using the social network analysis method, and explored the clustering and degrees of centrality characteristics of cultural memes. We then analyzed the similarities and differences among cultures of the dynasties. The main conclusions are as follows: (1) Cultural memes represent different cultural characteristics of dynasties, and the inheritance and differences among dynasties' cultures are closely related to their historical background. (2) Prevalence memes construct the cultural label of dynasties, which can roughly restore the cultural appearance of dynasties through fewer prevalence memes. (3) The use of community detection with a cultural meme network can determine the clustering of dynasties' cultures, and the degree of centrality further reflects the main cultural characteristics presented by successive dynasties.

**Keywords:** cultural meme; dynasty's culture; big data; social network analysis; word frequency analysis

## 1. Introduction

Culture is the material wealth and spiritual treasure created by a country in the course of its social and historical development. Its formation has a unique historical background, and throughout history, culture is inherited and developing. Unique culture took shape at different historical stages in China. After five thousand years of inheritance and blossoming, China has formed a rich and glorious history and culture, and also left a varied splendid cultural heritage. With the development of society and the acceleration of urbanization, people are paying increasing attention to the study of cultural heritage.

The study of culture cannot be separated from the previous definition of culture, and now there are more than 200 definitions of culture alone [1]. With the enrichment of research methods and data, cultural research has experienced both vertical and horizontal development, and gradually tends to become more specific and detailed. Vertical means to deepen the understanding of culture based on cultural research, and further analyze the cultural communication [2], evolution [3], memory [4], and space–time characteristics [5]. However, most researchers only consider the temporal and spatial characteristics of culture, and cannot comprehensively analyze it from multiple dimensions. Meanwhile, horizontally, the culture is gradually concretized into material culture [6,7], intangible culture [8,9],

regional culture [10], etc. Although the horizontal cultural studies have been detailed by degrees, the research on dynasty culture is still lacking.

With the continuous development of digital technology, the management of museum cultural relics has gradually entered the digital era, breaking the previous studies on the practical significance, role [11], management [12], and how to reflect regional culture [13] of museums as a cultural heritage warehouse [14]. With the advent of the big data era, data acquisition is no longer an obstacle. Windhager [15] analyzed a variety of methods to visualize cultural heritage through digital means so that people could visually understand cultural heritage better. Through the visualization of cultural heritage, rich image data can be obtained. Briola [16] provides an automatic classification and management system for images, documents, and other data, saving the cost of data cleaning and accelerating the acquisition of effective data. With the popularity of visualization and automatic classification methods, it becomes more convenient for people to obtain cultural heritage data. It also provides support for quantifying culture effectively. Research data also tend to a fusion of multi-source data from text, pictures, and sounds.

The quantification of culture has always been a key point in cultural studies. With the rapid development of computer technology, the application of computer technology [17–19] to cultural studies has provided scholars with many new methods and perspectives. Although the quantitative analysis of culture achieved some success, it is still short of a more granular perspective. Richard Dawkins published in the book "The Selfish Gene" [20] that culture is composed of memes. Memes are defined as a cultural unit that can be shared and passed down in different periods. They resemble the genes in biology, through which culture could be quantified in a fine-grained way.

Therefore, this paper employs the social network method to integrate multi-dimensional cultural characteristics, constructing relationships between multiple objects, in order to understand the structural characteristics of the culture. Unlike previous studies which only concentrated on space–time characteristics horizontally and focused on one single research object, the diversity of cultural characteristics is fully considered in our method from the perspective of cultural memes, providing a new method for cultural study. So, this paper mainly uses semantic word segmentation and word frequency statistics to extract dynasty cultural memes from the names in different dynasties, in order to, explores the cultural characteristics of dynasties from a more granular perspective. With the combination of quantitative methods in social network analysis [21–23]. This paper explains the cultural characteristics of the dynasties and discusses the similarities and differences between the cultures of different dynasties from the perspective of cultural meme networks.

The structure of this paper is as follows: Section 2 describes the related work referred to in this study. Section 3 describes the concepts, research framework, and methodology in this paper, including cultural meme extraction, cultural meme classification, the Louvain community detection algorithm, and degree centrality analysis. Section 4 performs the temporal analysis for cultural memetic characteristics and analyzes the similar clusters of dynasties in cultural memetic networks, as well as the structural characteristics of dynasties. Section 5 is the conclusion and discussion of the work.

## 2. Relation Work

With the vigorous development of computer technology, the combination of computer and culture has become an irresistible trend, enabling analyzing culture quantitatively from a different angle. For example, Suaib [24] combined computer graphics and media science to protect Malaysia's cultural heritage, transcending the traditional measures which analyzed the cultural heritage theoretically. Gossa et al. [17] analyzed West African folk tales using natural language processing and generated new story structures, and further explored information such as cross-cultural differences in narrative structures in combination with machine learning. Similarly, since a single type of data can only provide a partial analysis of cultural characteristics, Liu [18] effectively combined ancient Chinese history and culture with the knowledge map, not only allowing us to understand Chinese history and culture more conveniently but also coalesce a fusion method of multi-source data with cultural study. Jankovi [19]

classified cultural heritage through several different image classification methods, which not only provided better suggestions for museum management and cultural heritage preservation but also facilitated people's convenient access to multi-source cultural data.

Although the booming of computer technology has gradually enriched cultural research, there is still a lack of more fine-grained analysis methods. Nonetheless, memes have solved this problem well. The early research on memes mainly focused on the conceptual qualitative discussion of different fields. In the Oxford English Dictionary [25], a meme refers to an element that can be passed from culture to culture by imitation or other non-genetic means. Heylighen [26] believed that cultural features are similar to genes or viruses, while a meme is the unit of cultural imitation or communication. With the advancement of research, it is found that memes are a new cultural information unit, which can complete cultural evolution through reproduction, transmission, mutation, and natural selection just like the evolution of biological genes [3,27,28]. Among them, Henrich [3] believed that our culture is transmitted among individuals and passed down from generation to generation through social learning, and expounded the adaptation model of its cultural evolution from the evolution of the psychological mechanism underlying human social learning and the evolutionary dynamics of the cultural system. Mundinger [27] started from animal culture, analyzing the related theories of human culture and non-human culture, and elaborated on them from three aspects: micro-culture, cultural evolution, and the evolution of cultural evolution. Heylighen [28] divided the selection of cultural memes into four different stages: assimilation, retention, expression, and dissemination, and discussed the standards of cultural evolution. The above analysis on the definition of memes illustrates that culture can actually be broken down into individual memes.

Although culture can be made up of memes, how to extract memes and how to quantify memes are still questions under exploration. With the explosive growth of data, memes can be extracted from the text. Malhotra [29] extracts cultural memes from "tort stories" through data collection, and uses them to explore the impact on attitudes towards infringement reforms. Yong-Jun [30] extracted trademark names based on memetics to discuss cultural memes and further understand their cultural composition. Shin [31] believed that the film label was a brief description of film characteristics, and the extraction of film memes through movie labels was similar to the inheritance and variation of biological genes, and movie memes also had their specific rise and fall. Memes can not only be extracted from text, but also from pictures. Theisen [32] adopted a visual recognition pipeline to automatically discover political meme types with different appearances and explored the use of election pictures with special backgrounds to extract political meme types.

## 3. Data and Methods

### 3.1. Data Sources and Processing

The data in this paper are the catalog data of cultural relics in the collections published by the State Administration of Cultural Heritage. The data set contains a total of more than 2 million pieces of cultural relics, mainly including the name, category, dynasty, and other basic attributes of cultural relics, and the data lacking key information are cleaned. According to the Measures for the Administration of Museums issued in 1986, the names of historical relics are composed of three parts, namely, age, style or author, texture or color, and shape or application. The name of each type of cultural relic has a standard pattern, which mainly includes the characteristics of a dynasty, material, shape, texture, inscription, glaze color, shape, kiln, etc. The names of cultural relics not only reflect the main features of the collection, but also the most direct embodiment of the culture of dynasties. Therefore, this paper selects the data of the two attributes of dynasties and cultural relic names. However, due to the continuous updating of the digitization process of the National Collection Museum, the data adopted in this paper have a certain integrity deviation, which cannot contain all the cultural relic information of each dynasty. Secondly, with the passage of time, the number of preserved cultural relics is greatly reduced, so the digitized information of cultural relics is not complete. Although the data have a

certain deviation, it does not affect the actual physical meaning behind the data. The names of the dynasties involved in the pictures in this paper have adopted the abbreviations in Table A1.

*3.2. Relevant Concepts*

3.2.1. Cultural Meme

A cultural meme refers to a single word or phrase describing specific characteristics of relics obtained from word segmentation of the relic's names. It is the smallest unit constituting the culture of a dynasty, which, like a biological gene, is inherited. The inheritance and variability of cultural memes indirectly reflect the similarities and differences of dynasties' cultures. During this period, environmental changes will cause cultural memes to mutate like biological memes, leading to cultural differences.

3.2.2. Cultural Meme Types

Cultural meme types refer to a classification of cultural memes based on the naming rules of relics. There are ten different types, including color, texture, location, auxiliary, character, shape, dermatoglyphic pattern, use, craft, and others. Among them, color descriptors are called "color memes"; words that can be used for auxiliary decoration and expression of cultural relics are called "auxiliary memes"; words that describe various patterns on cultural relics are defined as "texture memes"; and words describing the shape of cultural relics are defined as "shape memes".

3.2.3. Prevalence Memes

Prevalence memes refer to the top three most common cultural memes of different meme types in each dynasty. For example, prevalence memes of color in the Stone Age are the three most frequently occurring colors of that age. Through them, the cultural face of the dynasty can be roughly reproduced, which provides great help for us to understand the culture of a dynasty.

*3.3. Research Framework*

The research framework for this paper consists of four main parts: Data processing, cultural meme extraction, cultural meme classification, and cultural meme analysis, as shown in Figure 1.

First, the purpose of data processing is mainly to classify the data into 20 dynasties according to the attributes of the dynasties they belong to. Figure 1A shows the process of grouping all Stone Age relics together. Second, specific cultural memes were extracted by word segmentation and word frequency statistics through the Jieba database. Cultural memes were classified, expressing the subordinate relationships among cultural memes, types, and prevalence memes. Finally, cultural memes were analyzed from both a time sequence aspect and network structure aspect to indicate cultural characteristics in each dynasty. The figures in this paper are chronologically listed.

*3.4. Method*

3.4.1. Cultural Meme Extraction

In this paper, the Jieba database [33] is used to classify the names of cultural relics in each dynasty, because the Jieba database can process a short text into words or phrases with the smallest unit granularity, which is a better way to obtain cultural memes. After grouping the cultural relic data according to the dynasties, the Jieba database is used to segment the cultural relic names of each dynasty. As for the Stone Age shown in Figure 1B, each relic name is segmented at first, excluding the common words in Chinese. After that, words and phrases are sorted based on their occurrence frequency. Thus, the top 1000 words or phrases in each dynasty are obtained. Then, the word frequency statistics are carried out. After removing the words with a word frequency of 1, synonyms are merged, and the 1050 most frequent cultural memes are selected, forming a meme matrix of $20 \times 1050$, as shown

in Table 1. The abscissa is the extracted 1050 cultural memes, and the corresponding value is the word frequency in the corresponding dynasty.

   After obtaining the cultural meme matrix, memes are classified into different cultural meme types, shown in Figure 1C. Among all the types, color, auxiliary dermatoglyphic pattern, and shape are relatively intuitive cultural relic characteristics. Therefore, this paper mainly elaborates on these four memes types. On the basis of cultural meme types, the top three popular cultural memes in different cultural meme types in each dynasty are obtained according to the definition of prevalence memes.

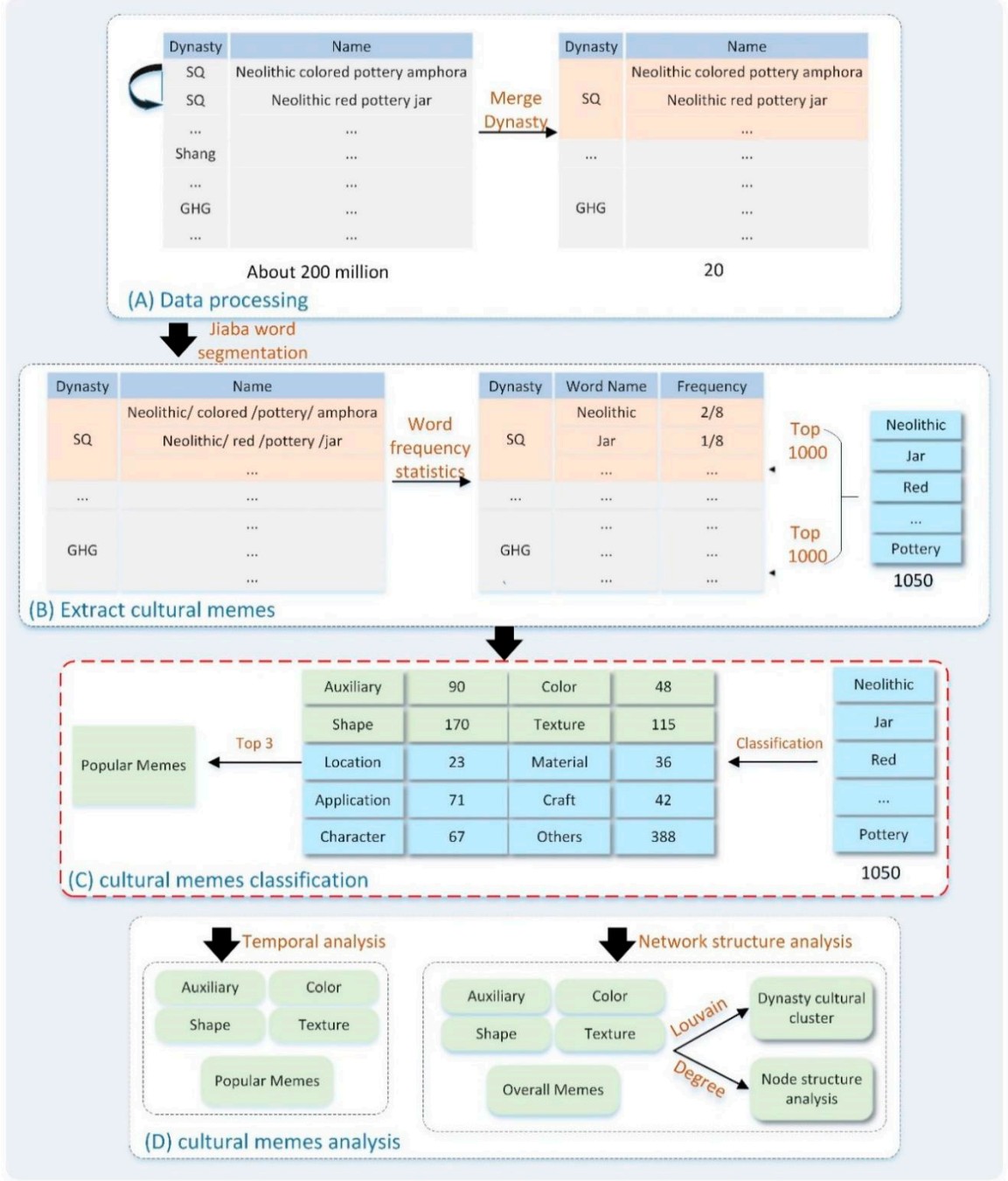

**Figure 1.** Research framework. (**A**) Data processing; (**B**) Extract cultural memes; (**C**) Cultural memes classification; (**D**) Cultural memes analysis.

**Table 1.** Vector matrix of cultural memes in dynasties.

| Dynasty | Copper | KuiLong | Yellow | ... | Binaural | Accessories | Hollow Out |
|---------|--------|---------|--------|-----|----------|-------------|------------|
| SQ | 0.000105 | | 0.000192 | ... | 0.014313 | 0.000051 | 0.000158 |
| Shang | 0.041962 | 0.001783 | 0.000047 | ... | 0.001821 | 0.00049 | 0.00033 |
| Zhou | 0.103191 | 0.001135 | 0.000138 | ... | 0.001607 | 0.000634 | 0.000744 |
| | | | ... | | | | |
| Qing | 0.012381 | 0.00135 | 0.000941 | ... | 0.001338 | 0.000311 | 0.000303 |
| MG | 0.004657 | 0 | 0.000213 | ... | 0.000239 | 0 | 0.00018 |
| GHG | 0.004407 | 0 | 0.000323 | ... | 0 | 0 | 0 |

### 3.4.2. Louvain Community Detection Algorithm

The Louvain algorithm [34,35] is a community detection algorithm based on modularity, which can discover hierarchical community structure and optimize it by maximizing the modularity of the entire community network. The algorithm first takes each point in the network as a community, calculating the module-degree increment of every neighborhood node in the community, in order to find out the point with the maximum module-degree increment and to merge into a new community. Then a new community is taken as a point, and the process above is repeated until the results no longer change.

The Louvain algorithm can quickly and efficiently discover the characteristics of community clusters, detecting the similarities and differences between cultures of different dynasties from the perspective of cultural memes, in order to obtain the dynasty cultural clusters. This paper not only analyzes the temporal characteristics of cultural memes but also calculates the Pearson similarity between the four types of cultural meme matrix vectors with dynasty for the node. The dynasties belonging to the same cluster have strong similarities in cultural characteristics. Through community detection of different types of cultural meme networks, the similarity and difference of cultural characteristics between dynasties can be obtained.

### 3.4.3. Degree of Centrality Analysis

The degree of centrality is a measure of the network node importance in cultural meme networks, and a dynasty centricity node degree is higher when connected with a dynasty node. In the cultural memetic network, the higher the centrality of a certain dynasty node, the more dynasty nodes are connected to it, and the more similarity with other dynasty cultures. When lower, it means that fewer dynasty nodes are connected to it and the culture of other dynasties mainly showed differences. The main calculation formula was as follows:

$$D(i) = \sum_{j=1}^{n} a_{ij}$$

where $D(i)$ denotes the degree of centrality of nodes $i$, $a_{ij}$ denotes the adjacency matrix of nodes in the network, and the meme network of the same type establishes the connection relationship between nodes in dynasties by selecting the appropriate threshold size. If the value is greater than the threshold, the value is 1; if the value is below the threshold, the value is 0.

## 4. Analysis of Cultural Characteristics of Dynasties

### 4.1. Timing Analysis of Cultural Meme Characteristics

#### 4.1.1. Color Memes

The color meme is a special cultural phenomenon and a unique cultural meme, composed of the frequency of words related to color. The symbolic meaning of color has existed since ancient times, and it is generated through the screening of long-term historical development. There are two kinds of colors: positive colors and negative colors. Positive colors are also called the "five colors", green, red,

yellow, white, and black. Generally speaking, the positive colors are positive, affirmative, and upward, while the negative colors are negative, passive, and low. The "five colors" correspond to the five elements: Water = black, fire = red, earth = yellow, wood = green, and gold = white.

By combining the colors of the same color system, this paper summarizes the ten colors of black, white, red, green, blue, yellow, tricolor, brown, verdant, and gray to obtain the colors of cultural relics of the dynasty. Among them, black, white, red, green, and yellow are the basic "five colors", while verdant, blue, brown, gray, and tricolor are special combination colors. Figure 2 shows the peripheral area showing the corresponding dynasty, the lines of each color represent different color memes, and the point corresponding to the radius is the proportion of the color meme in the cultural meme of the dynasty.

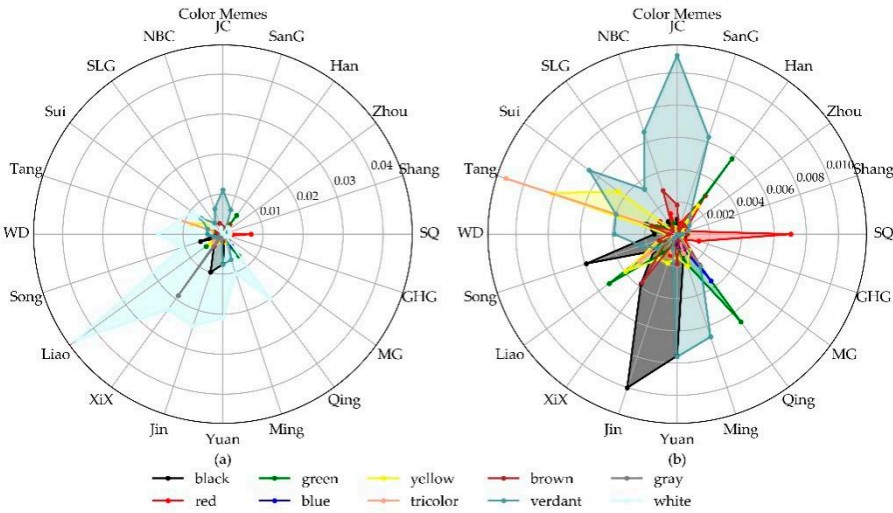

**Figure 2.** Color meme distribution trend. (**a**) Shows the temporal distribution of the ten color memes, and (**b**) Shows the temporal distribution of the two colors after removing gray and white.

Figure 2a shows that white and verdant memes have appeared in all dynasties, reflecting the inheritance of color memes. The proportion of white meme was much higher than other color memes from the Sui dynasty to the Qing dynasty and reached its peak in the Liao dynasty. It was the most prevalent color meme in the Liao dynasty. Verdant was prevalent from the "period of the Three Kingdoms" to the Ming dynasty. However, it experienced a downturn from the "period of the Five Dynasties and Ten Kingdoms" to the Jin dynasty, which may be related to the popularity of black memes in this period. The red meme in the Stone Age is much more common than in other dynasties, as the red meme has always been a favored color in ancient times. In the Stone Age, people extracted pigments from hematite and used them on bone utensils, stone tools, and pottery. Due to the famous tricolor of the Tang dynasty, the tricolor meme in the Tang dynasty is much more common than in other dynasties, which shows the different characteristics of the color meme.

Since the "five colors" correspond to five virtues, we obtained the attributes of the five virtues advocated in different dynasties through searching for literature and collected the top three color memes of the dynasty. Among them, the attributes of five virtues in the Stone Age were missing due to insufficient data. Shown in Table 2, the top three color memes of each dynasty are basically matched to the "five colors'" attributes corresponding to the five virtues, indicating that the color memes extracted from the names of cultural relics can reflect the color culture and cultural meme characteristics of the dynasties.

**Table 2.** Attributes of five virtues and the corresponding five colors in dynasties.

| Dynasty | Five Virtue Attributes | Five Color Attributes | Top Three Colors | Dynasty | Five Virtue Attributes | Five Color Attributes | Top Three Colors |
| --- | --- | --- | --- | --- | --- | --- | --- |
| SQ | | Red Black White | Red White Yellow | Tang | Earth Fire | Yellow Red | White Tricolor Yellow |
| Shang | Gold | White | White Verdant Red | WD | Gold Earth Water Wood | White Yellow Black Verdant | White Verdant Black |
| Zhou | Fire | Red | White Verdant Red | Song | Fire | Red | White Black Verdant |
| Han | Water Soil Fire | Black Yellow Red | Green Red Yellow | Liao | Water | Black | White Green Yellow |
| SG | Soil Fire | Yellow Red | Verdant Red Brown | Jin | Gold Earth | White Yellow | White Black Yellow |
| JinC | Gold | White | Verdant Brown Black | Yuan | Gold | White | White Verdant Black |
| NBC | Water Wood Fire | Black Verdant Red | Verdant Brown White | Ming | Fire | Red | White Verdant Yellow |
| SLG | Fire Water Wood Gold | Red Black Verdant White | Verdant Black Gray | Qing | Water | Black | White Green Blue |
| Sui | Fire | Red | White Verdant Yellow | | | | |

### 4.1.2. Texture Memes

Texture memes can often be divided into a real-life category, imaginary auspicious birds and animals category, and religion category. As shown in Figure 3, the real-life texture memes are represented by small dots, the imaginary auspicious birds and animals are represented by crosses, and the religious category is represented by triangles. Among them, grate pattern, cloth pattern, arc pattern, flower pattern, ring pattern, curve pattern, and basket pattern belong to the category of real life, while lotus pattern and tangled lotus pattern belong to the category of religion. Dragon pattern, double phoenix pattern, double dragon patterns, and feather pattern belong to the category of imaginary auspicious birds and animals. Since the dragon pattern meme was very popular in the Qing dynasty, its value was much higher than that of other dynasties. This is because the imperial power was highly concentrated in the Ming dynasty and Qing dynasty, and the dragon pattern was solely owned by the family.

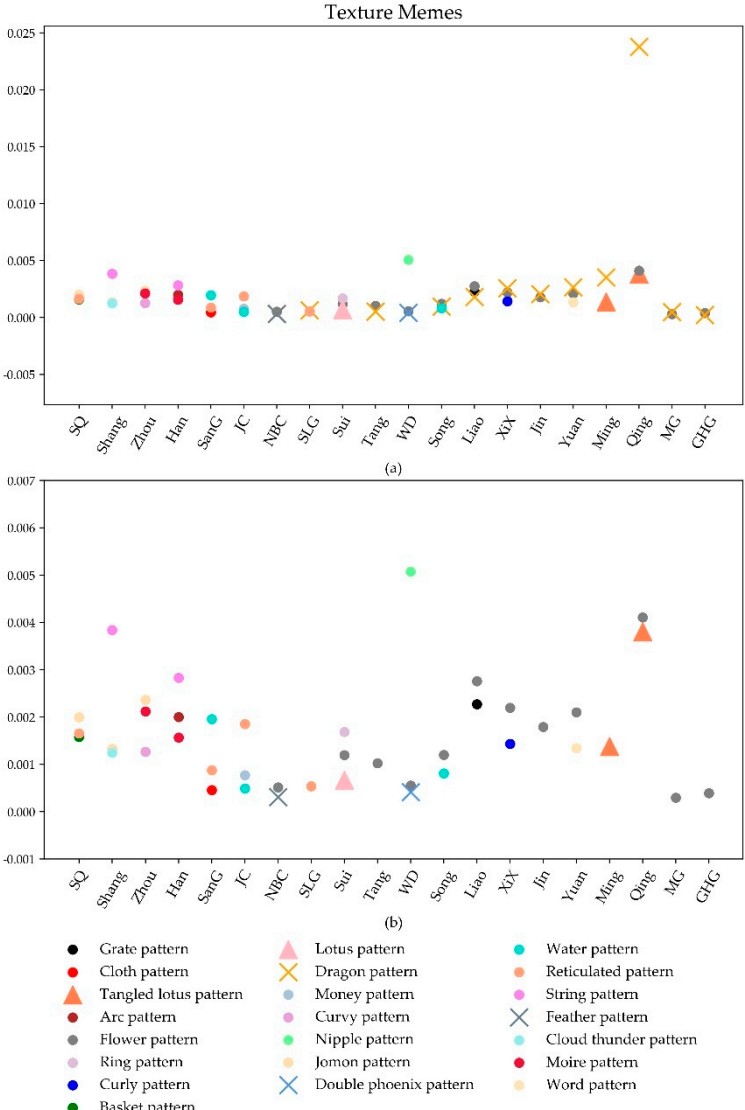

**Figure 3.** Distribution trend of texture memes. (**a**) Shows the top three texture memes in each dynasty. (**b**) Shows the texture meme except for the dragon pattern.

In Figure 3, the dragon pattern, double dragon pattern, water pattern, reticulated pattern, and other real-life patterns and imaginary auspicious bird and animal patterns appeared more frequently. Among all the top three decorative memes in each dynasty, the real-life category of

texture memes occupies the majority. However, the real-life texture memes had a transformation process from plain life in the early stage to the spiritual pursuit in the later stage. From the Stone Age to the "Jurchen Jin dynasty", the jomon pattern, reticulated pattern, water pattern, and other simple texture patterns were the main types in the real-life category, with the famous "Jomon culture". After the Sui dynasty, the occurrence frequency of flower patterns gradually became prominent, and people started expressing their emotions with styles in real life, focusing on spiritual expression. As religious ornamentals, the tangled lotus pattern and lotus pattern originated in the Han dynasty and gradually prevailed in the "Northern and Southern dynasties", Sui and Tang dynasties, and Song and Yuan dynasties, reflecting the inheritance of decorative memes. The cloud thunder pattern mainly appears on bronzes, which began in the late Stone Age period. In the early Shang dynasty, white pottery, hard pottery, and primitive celadon were made in the Shang and Zhou dynasties. The cloud thunder pattern is the ground pattern that sets off the main pattern and reflects the difference of the pattern memes.

### 4.1.3. Auxiliary Memes

Figure 4 shows the spatial and temporal distribution of auxiliary memes, with the x-axis representing the dynasties and the y-axis representing the proportion of auxiliary memes in all memes for each dynasty. The temporal changes of animal-assisted memes reflect people's pursuit and attitude towards the legend of gods, imperial power, lifestyle, etc., as shown in the red box of Figure 4a. Common animal memes, such as tiger, chicken, cattle, and horse, accounted for a large proportion in all dynasties. The high occurrence frequency of Suzaku, white tiger, unicorn, gluttonous, and basalt are related to people's belief in legend and gods. The proportion of dragon and phoenix in the Ming and Qing dynasties was higher than that in other dynasties, which reflected the highly concentrated pursuit of imperial power. During the Stone Age and Shang dynasty, mussels, seashells, turtles, and beasts were more common. The Shang dynasty used to record narration on the carriers of animals, shellfish, and tortoises, which reflected people's special lifestyle. Auxiliary memes of flowers and plants highlight people's pursuit and attitude at the spiritual level, as shown in the red box in Figure 4b. In earlier periods, such as the Stone Age, Shang dynasty, and Zhou dynasty, people mostly adopted lute, speaker, banana leaf, gourd, and other common things in life, while in later periods, people often used bright moon and flowers to express their spirit and emotions. Similar to the real-life memes in texture, they have undergone a change from the pursuit of simple living in the early stage to the focus on the spiritual level in the later stage.

### 4.1.4. Shape Memes

The shape of cultural relics can usually be divided into mouth shape, ear shape, hole shape, foot shape, neck shape, and belly shape. The temporal distribution of these memes is shown in Figure 5. From Figure 5a, it is obvious that the ear shapes of cultural relics are "four series", "dual line", "binaural", and "single ear". In Figure 5b, "tripod", "square foot", and "high foot" were popular in previous dynasties. The neck shape was dominated by "long neck"; "bulging", "deep belly", and "arc belly" occupied an important position in the dynasties. In Figure 5c,d, "square", "round", "rectangle", and "flat" are all the most popular artifact forms in previous dynasties, which all show the inheritance of shape memes. However, "lick", "exposure", and "Huakou" were particularly prominent in the "period of the Five Dynasties and Ten Kingdoms", and Song and Liao dynasties. The Stone Age cultural relics were mainly "binaural", the Han dynasty bronze ware commonly featured the "four beasts" and "eight beasts", the Stone Age and Liao dynasty bronze ware commonly featured the "single hole", and in the "period of the Five Dynasties and Ten Kingdoms", the "Western Xia", and Ming dynasty bronze wares, the "square hole" was prevalent, which all showed that the utensil-like memes had the different characteristics of dynasties.

### 4.1.5. Prevalence Memes

In this paper, the average value of meme vectors of all dynasties in each meme type matrix is calculated respectively to find the most prevalent meme types in each dynasty. Figure 6 shows the nine meme types classified in this paper except for the other categories. It can be seen in Figure 6b that every dynasty concentrates on memes of craft, texture, and application. Good texture and craft are the cornerstones for creating rich and colorful cultural relics, while practical uses are the driving force for the continuous generational use of cultural relics. In Figure 6a, from the Tang dynasty to the Qing dynasty, the popularity of color memes was significantly higher than it was during the previous dynasties. On the whole, texture memes showed a U-shaped distribution trend, with slightly higher ends and lower middle.

The most prevalent culture meme of each dynasty can be found through the distribution of the popular trend of different types of memes. In Figure 6a, the color meme of the Tang dynasty and Liao dynasty is more prominent than that of other dynasties due to the tricolor of the Tang dynasty and white porcelain of the Liao dynasty. So, the color culture of the Tang dynasty and Liao dynasty is more symbolic. The auxiliary memes of the Shang dynasty, Tang dynasty, Liao dynasty, and Qing dynasty are also more prominent. Texture memes and shape memes were particularly popular in the Zhou dynasty, Han dynasty, and Qing dynasty. During the period of the "Republic of China" and "People's Republic of China", due to the impact of the war, there will be more local memes.

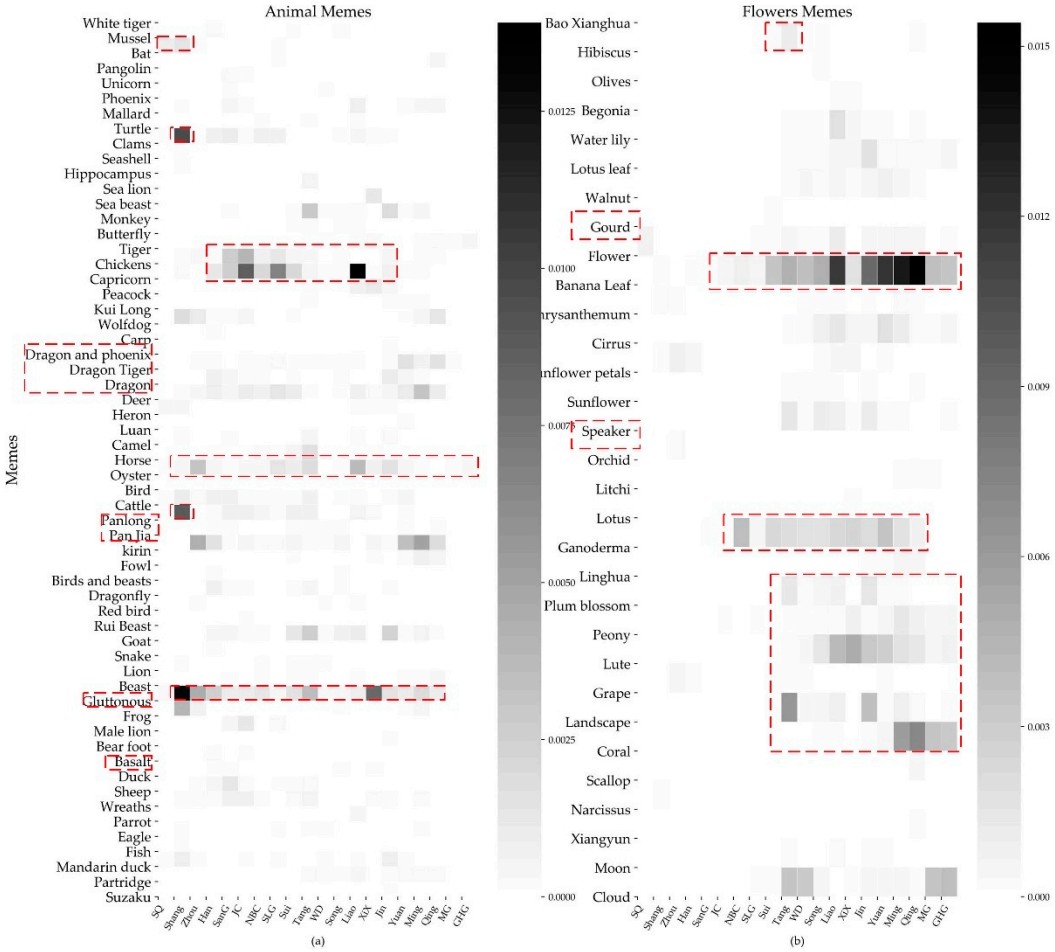

**Figure 4.** Distribution trend of auxiliary memes. (**a**) Illustrates the meme related to animals, while (**b**) is about floristic memes.

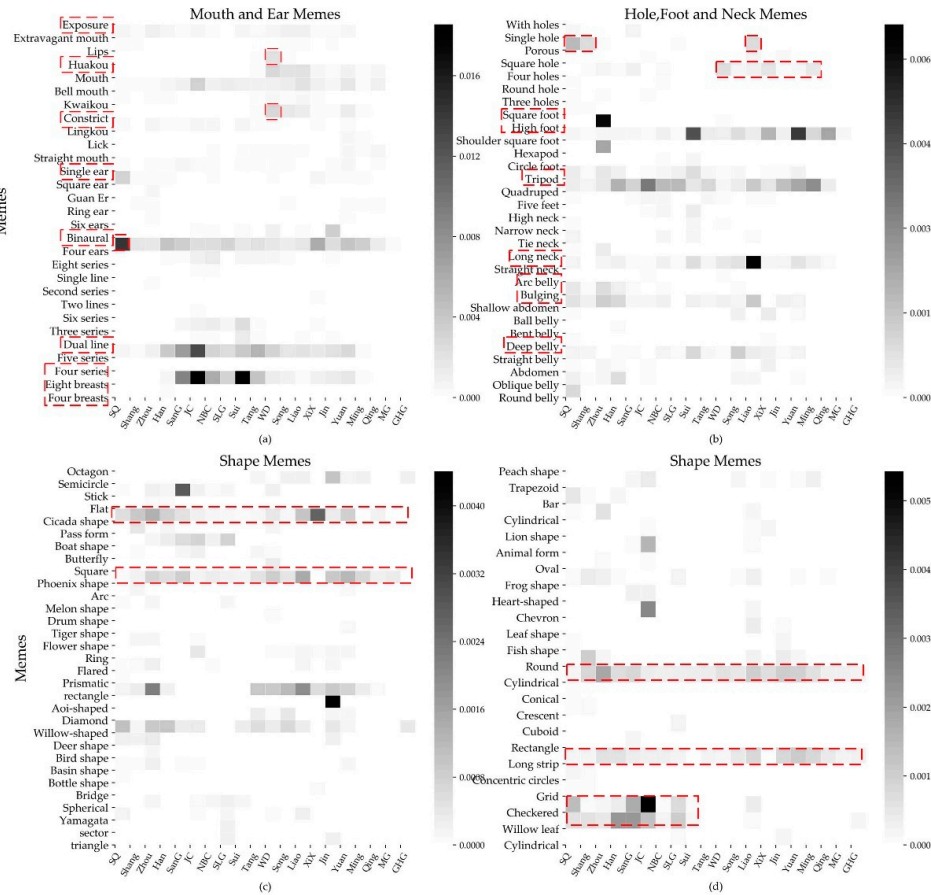

**Figure 5.** Distribution trend of shape memes. (**a**) Shows the distribution of mouth shape and ear shape. (**b**) Shows the distribution of hole shape, foot shape, and neck shape. (**c**,**d**) are the overall shape description of cultural relics.

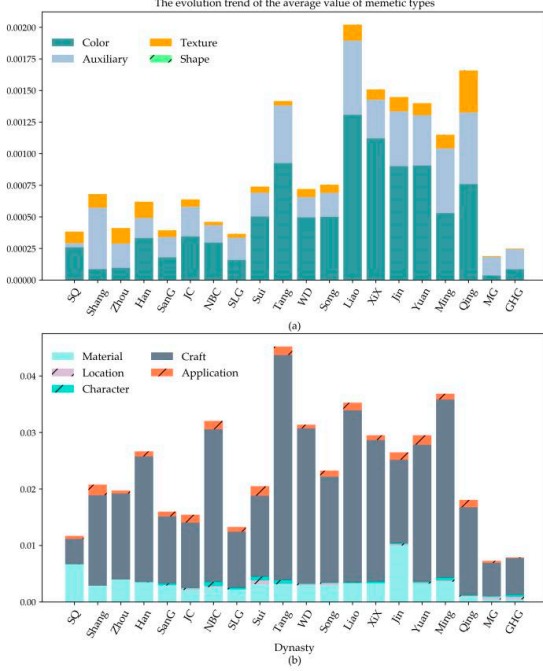

**Figure 6.** Distribution trend of meme types. (**a**) Shows the distribution trend of the four meme types: color, texture, auxiliary, and shape, and (**b**) Shows the distribution trend of the five meme types: material, craft, location, application, and character.

Here, the top three memes of each meme type in each dynasty were counted to construct a unique cultural prevalence meme label representing each dynasty's history and culture (Figure 7). In terms of materials used, copper, pottery, and porcelain were the most common material memes in all dynasties, and stone and bone were popular in a few periods such as the Stone Age and Shang dynasty. Hongzhou, Yaozhou, and Jingdezhen were mostly found in the location memes of the dynasties, and these three places were all popular for the production of ceramics. Most of the application memes in all dynasties were based on common articles of daily life, such as bowls, jars, and figurines, while arrows and spears were produced in large quantities in the Zhou dynasty due to the frequent wars in this period.

| Dynasty | Material | Location | Auxiliary | Character | Craft | Texture | Shape | Color | Application |
|---|---|---|---|---|---|---|---|---|---|
| SQ | Stone Pottery Bone | | Mussel Flower Deer | Deity | Engraved Wear Painted | Jomon Reticulated Basket pattern | Binaural Double Concave | Red White Brown | Jar Adze Device |
| Shang | Copper Bone Jade | | Beast Turtle Cow | | Engraved Uncolored Face Tie beam | String pattern Jomon Cloud thunder pattern | Double Binaural Triangle | White Verdant Red | Bu Sickle Spear |
| Zhou | Copper Pottery Jade | | Beast Pan Jia horse | Sima | Back wear Uncolored Face Painted | Jomon Moire Curvy | Square foot Triangular Double | White Verdant Red brown | Arrow Spear Hook |
| Han | Copper Pottery Watt | | Beast Pan Jia Chicken | Deity Sima Female | Painted Sealing mud Wear | String pattern Moire Arc | Short Binaural Dual line | Green Red White | Glaze Jar Figurines |
| SanG | Copper Pottery Porcelain | Hongzhou Wuzhou | Tiger Chicken Duck | Deity Sima Buddha | Back wear Pile up Uncolored Face | Water ripple Reticulated Cloth pattern | Four series Dual line Binaural | Verdant Red White | Glaze Bowl Jar |
| JC | Porcelain Copper Pottery | Hongzhou Wuzhou | Chicken Frog Beast | Male Deity Samurai | Stippling Painted Printing | Reticulated Money pattern Water ripple | Four series Dual line Grid | Verdant Black Brown | Jar Glaze Bowl |
| NBC | Pottery Copper Bone | Hongzhou Wuzhou | Lotus Chicken Beast | Buddha Samurai Bodhisattva | Painted Back wear Deposit | Pattern Dragon pattern Feather pattern | Four series Stand up Dual line | Verdant Brown White | Figurines Statue Bowl |
| SLG | Copper Porcelain Pottery | | Chicken Dragon Horse | General Captain Hou | Back wear Painted Stippling | Dragon pattern Water ripple Reticulated | Four series Binaural Dual line | Verdant Brown Verdant Yellow | Jar Tomb Glaze |
| Sui | Copper Porcelain Pottery | Hongzhou Wuzhou | Flower Lotus Chicken | Guanyin Bodhisattva Buddha | Painted Printing Engraved | Ring pattern Flower pattern Lotus pattern | Ride Stand up Dual line | White Verdant Yellow | Glaze Statue Figurines |
| Tang | Copper Pottery Porcelain | Hongzhou Yaozhou | Grape Flower Beast | Female Buddha Samurai | Painted Back wear Wear | Flower pattern Double Dragon Pattern Dragon pattern | Ride Stand up Dual line | Tricolor White Yellow | Glaze Figurines Tomb |
| WD | Copper Porcelain Pottery | Quzhou Yaozhou | Flower Moon Lotus | Male Buddha Lux | Back wear Gold-plating wear | Nipple pattern Flower pattern Double phoenix pattern | Bare back Hold Dual line | White Verdant Black | Glaze Bowl Tomb |
| Song | Copper Porcelain Pottery | Jingdezhen Huzhou | Flower Lotus Peony | Samurai Male Buddha | Engraved Printing Back wear | Flower pattern Dragon pattern Water ripple | Hold Huakou Dual line | White Black Verdant | Glaze Bowl Cup |
| Liao | Copper Pottery Gilt | Jingdezhen | Chicken Flower Peony | Buddha Boy Guanyin | Engraved Painted Relief | Flower pattern Grate Dragon pattern | Long neck Bare back Hold | White Green Tricolor | Glaze Bottle Bowl |
| XiX | Copper Porcelain Sandstone | | Beast Peony Lotus | Buddha Bodhisattva Lux | Painted Woodcarving Engraved | Dragon pattern Flower pattern Curly Pattern | Bare back Binaural Flat | White Gray Brown | Glaze Bowl Jar |
| Jin | Gold Copper Porcelain | Yaozhou Hebei | Flower Grape Peony | Boy Priest | Engraved Painted Printing | Double Dragon Pattern Dragon pattern Flower pattern | Double Bare back Rectangle | White Black Red green | Glaze Bowl Jar |
| Yuan | Porcelain Copper Jade | Jingdezhen Yaozhou | Flower Lotus Pan Jia | Buddha Bodhisattva Shakyamuni | Engraved Printing Back wear | Dragon pattern Flower pattern Word pattern | High foot Fold Binaural | White Black Verdant | Glaze Bowl Jar |
| Ming | Copper Porcelain Gilt | Jingdezhen Beijing | FLower Scape Pan Jia | Buddha Bodhisattva Guanyin | Engraved Back wear Painted | Dragon pattern Tangled lotus pattern Double Dragon Pattern | Binaural Tripod Stand up | White Verdant Green | Glaze Jar Bowl |
| Qing | Copper Porcelain Gilt | Jingdezhen Jilin | Flower Cloud Scape | Buddha Bodhisattva Guanyin | Gold drawing Back wear Carving | Dragon pattern Flower pattern Tangled lotus pattern | Fold High foot Binaural | White Blue Green color | Bowl Plate Glaze |
| MG | Copper Stone Porcelain | Beijing Shandong | Moon Flower Scape | Lu Xun Red Army | Engraved Back Wear Woodcut | Dragon pattern Flower pattern | Stand up Fold Square | White Red Yellow | Tomb Statue Monument |
| GHG | Wood Copper Silver | Henan Nanjing | Flower Moon Scape | Mao Zedong Educated youth Female | Brocade Back wear Batik | Flower pattern Dragon pattern | Stand up Lace Diamond | Red White Black | Tomb Mirror Gun |

**Figure 7.** Prevalence meme labels in all dynasties.

Through the analysis of the meme labels of the popular culture of each dynasty, the general historical and cultural features of that dynasty can be represented. The Stone Age was a time of the worship of gods. Stone, pottery, and bone were usually used to create binaural and single ear cultural relics using carving and painting techniques, such as jars, adze, and devices. People liked jomon, reticulated, mussel, flower, and deer memes. The colors were mostly red, white, and brown. The Song dynasty advocated Buddhism and paid attention to military force. It used engraving,

printing, and other crafts to produce bowls and lamps decorated with patterns, dragon patterns, peony, and lotus on the texture of copper, porcelain, and pottery. Jingdezhen ceramics were very famous. In addition to the use of ceramics, the gilt was popular during the Qing dynasty, the use of which was mostly intended to depict the gold decoration. Blue and green colors were used to make the cultural relics more beautiful and a large number of dragon and flower patterns, landscapes, and other ornaments also stressed the concentration of the imperial power and the pursuit of spirituality.

*4.2. Analysis of the Characteristic Structure of Cultural Memes*

4.2.1. Clustering of Dynasties' Cultures

This paper uses the Louvain community detection algorithm to obtain the cultural community clusters of dynasties in cultural meme networks, in order to analyze the cultural characteristics for each dynasty. According to the cultural meme types classified in this paper, the overall network, color network, texture network, shape network, and auxiliary network are generated. The threshold of the color network is 0.9, while it is 0.6 for other networks. Nodes in the networks represent the dynasty, while lines represent the Pearson similarity value of cultural memes between two dynasties. The size of the node is determined by the node degree—the larger the node is, the more important the dynasty is in the whole network.

As shown in Figure 8b, (1) the "period of the Three Kingdoms" to the Sui dynasty and Tang dynasty to the Ming dynasty in the overall network; (2) the "period of the Three Kingdoms" to the "period of the Sixteen States", the "period of the Five Dynasties and Ten Kingdoms" to the Qing dynasty in the color network; (3) the "Western Xia regime" to the Qing dynasty in the texture network; (4) the "period of the Three Kingdoms" to the Tang dynasty, the "period of the Five Dynasties and Ten Kingdoms" to the "Western Xia regime", the Yuan Dynasty to the "People's Republic of China" in the shape networks; (5) the "Northern and Southern Dynasties" to the Tang dynasty in the auxiliary network, all these show that the dynasties in one cluster are continuous. It indicates that each part near a dynasty will easily form a cluster, and the other dynasties' cultures show the characteristics of inheritance. In the overall network, during the time between the Tang dynasty and the Ming dynasty, because of the fragmentation of the countries in the "period of the Five Dynasties and Ten Kingdoms" and the conflict and melting between Zhong Yuan culture and Nüzhen culture in the "Jurchen Jin dynasty", dynasty cultures are different from each other. In the color network, the Shang dynasty, Zhou dynasty, Sui dynasty, and Ming dynasty belong to one cluster, sharing similar characteristics. However, color culture tends to alternate between dynasties, meaning that the cultural characteristics at an early stage will reappear after breaks. In the texture network, there is a great variation between the dynasties before the "period of the Five Dynasties and Ten Kingdoms". After that, the texture pattern became stabilized with evolution, showing similar characteristics between different dynasties. The continuity characteristic of the community cluster of the shape network is the most obvious, which shows the inheritance characteristic of the culture of all dynasties.

By analyzing the community clusters of different cultural meme networks, the similarities and differences among dynasties' cultures can be quantitatively assessed. The main purpose is to prove that the similarity of different cultures between dynasties will change due to cultural types. For example, from the "period of the Three Kingdoms" to the Tang dynasty, the overall network and the shape network basically belong to one cluster and, namely, from the perspective of the overall and shape networks, the similarity among dynasties' cultures is higher, while its auxiliary memes can be divided into two different clusters. The color and texture networks mainly highlighted cultural differences among the dynasties. By analyzing the similarities and differences between dynasties' cultures from the perspective of cultural memes, we can find out which types of cultural memes present similarities between dynasties and which present differences. The factors influencing the formation of a dynasty's culture are very complex and changeable. To elaborate from the perspective of cultural memes is conducive to a more detailed understanding of the similarities and differences among dynasties.

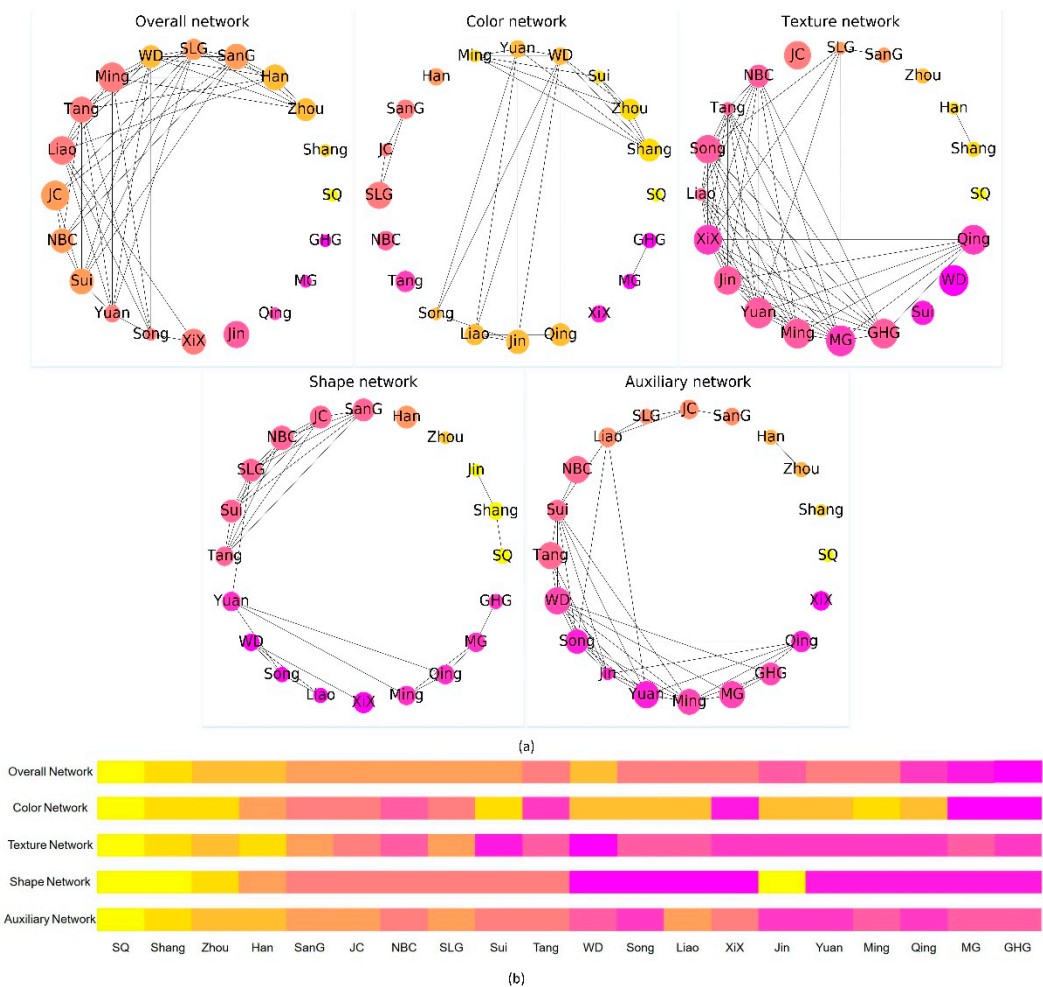

**Figure 8.** Cultural meme network community cluster. (**a**) Shows the connections of each network and the results of community detection. (**b**) Illustrates the temporal distribution of the community clusters for each dynasty.

### 4.2.2. Features of the Cultural Structure of Dynasties

The community clusters show that the cultures of dynasties in one cluster are similar to each other. The centrality can measure the level of similarity between different dynasty cultures. In Figure 9, red circles represent the dynasties with low centrality in networks, while blue boxes represent the dynasties with high centrality. The higher the centrality is, the more dynasties it connects, and the more important that dynasty is in the whole network, also showing the higher level of similarity it has. Low centrality illustrates less connected nodes, meaning this dynasty is unique from the others.

For the overall network, the distribution of node centrality shows a rising trend in general, indicating that over time there is an unceasing inheritance and fusion of cultures. However, there are some special turning points: The "period of the Three Kingdoms" to the Jin dynasty, the "period of the Sixteen States" to the Sui dynasty, the "period of the Five Dynasties and Ten Kingdoms" to the Song dynasty. These periods all experienced conversions from multiple rulers to one single ruler. The Liao dynasty was unique because of the conflict and fusion between its ethnic minority culture and the Central Plains Han culture. There are three main community clusters in the overall network. The Zhou dynasty, Han dynasty, and the "period of the Five Dynasties and Ten Kingdoms" belong to one cluster. the "period of the Three Kingdoms", the Jin dynasty, the "Northern and Southern dynasties", the "period of the Sixteen States", and the Sui dynasty belong to one cluster. The Tang dynasty, Song dynasty, Liao dynasty, the "Western Xia regime", Yuan dynasty, and Ming dynasty

belong to one cluster. The highest centrality of nodes in the network are the "period of the Five Dynasties and Ten Kingdoms", the "period of the Sixteen States", and the Ming dynasty, which belong to three different clusters, indicating that these three dynasties make the greatest contribution to the community cluster and have a higher level of similarity, mainly showing the cultural similarity with the same community cluster.

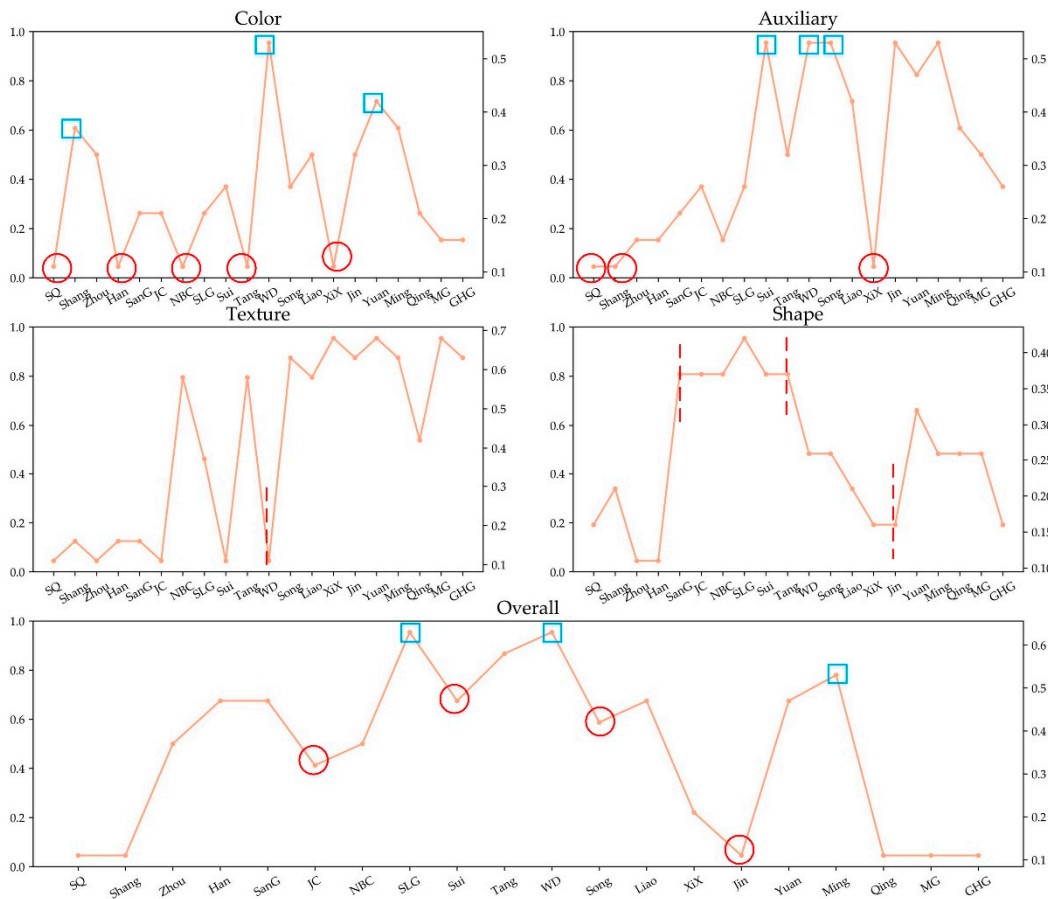

**Figure 9.** The distribution of centrality for nodes in cultural meme networks.

Similarly, through the analysis of centrality for nodes in other networks, cultural characteristics can be observed. In the color network, the Shang dynasty, the "period of the Five Dynasties and Ten Kingdoms", and the Yuan dynasty show similarities in color culture, while the Stone Age, Han dynasty, the "Northern and Southern dynasties", Tang dynasty, and the "Western Xia regime" are unique from the other dynasties. In the auxiliary network, the auxiliary cultures of the Sui dynasty, Tang dynasty, the "period of the Five Dynasties and Ten Kingdoms", and Song dynasty show similarities, while the Stone Age, Shang dynasty, and the "Western Xia regime" show uniqueness. In the texture network, dynasties at the early stage present different texture cultures, while the dynasties after the "period of the Five Dynasties and Ten Kingdoms" mainly presented similarities. The changing nodes in the shape network are the "period of the Three Kingdoms", Tang dynasty, and "Jurchen Jin dynasty", which also belong to three different community clusters from Figure 8b. The similarities of the shape culture for dynasties are mainly continuous.

## 5. Conclusions and Discussions

Based on the national cultural relic data released by the National Cultural Heritage Administration, we constructed the cultural memes of dynasties and used social network analysis to discuss the cultural

characteristics of dynasties and elaborate their historical and cultural features. The main conclusions are as follows:

1.  The temporal variation of cultural meme types expounds the cultural characteristics of the coexistence of inheritance and differences between the dynasties. Among them, the color meme reflects that the color culture of dynasties is closely related to the five virtues advocated by dynasties, verifying the possibility of reflecting the culture of the dynasty from the names of relics. Auxiliary memes and texture memes reflected the transformation of people's pursuit from simple life needs to spiritual development.

2.  By calculating the average value of cultural meme types of dynasties, it is found that craft, material, and application memes were very popular in all dynasties. After the Tang dynasty, color memes were more popular and abundant than it during the previous dynasties. Texture memes showed a U-shaped distribution trend on the whole, which represented the inheritance of prevalence memes in all dynasties.

3.  Statistical analysis of prevalence memes of successive dynasties helped to determine the popular cultural memes of a dynasty's culture and define the culture of the dynasties. To a certain extent, it helped us reproduce the cultural characteristics of the dynasties, which is conducive to a more comprehensive understanding of dynasty culture.

4.  The Louvain community detection algorithm was used to obtain the cultural similarity of clusters of dynasties in five different types of cultural meme networks. It was found that the cultural similarity of dynasties belonging to the same community cluster presented continuous characteristics.

5.  By analyzing the level of centrality of dynasty nodes in different networks, we could not only detect the similarity clusters of different culture types, but also find out the most important dynasty nodes within a cluster, and could judge the similarity and uniqueness of dynasty culture in different types of cultures.

From the perspective of the cultural meme, it is helpful for us to interpret the culture of dynasties by combining qualitative and quantitative methods, and to analyze the inheritance and variation in characteristics of the cultures of dynasties to develop a comprehensive understanding of the cultural facets of dynasties. Using the name data of cultural relics to construct cultural memes and studying the cultural characteristics of dynasties from a more fine-grained perspective provides a new way of thinking for future studies. However, this article only elaborates the dynasty culture from the perspective of a single cultural meme type and the overall cultural meme. The dynasty culture is not just a single body but is made up of several aspects, with cultures merging into new cultures. We hope to be able to continue this study and focus on the level of fusion between different cultures, carry out a more comprehensive analysis of the cultural characteristics of dynasties, and discuss the cultural causes of the differences. However, because the data obtained in this paper are not comprehensive enough and there are some errors, we cannot analyze the full view of the culture of the dynasty comprehensively.

**Author Contributions:** Methodology, Z.S.; Supervision, H.L., L.C., Z.C., S.L. and L.Z.; Writing—original draft, Z.S.; Writing—review & editing, Z.S. All authors have read and agreed to the published version of the manuscript.

**Funding:** This work was supported by the National Natural Science Foundation of China under Grant 41571397, Grant 41871364, Grant 41671357, Grant 41871302, and Grant 41871276. This work was carried out in part using computing resources at the High Performance Computing Platform of Central South University.

**Conflicts of Interest:** The authors declare no conflict of interest.

## Appendix A

**Table A1.** Abbreviation of dynasty name.

| Dynasty Name | Abbreviation | Dynasty Name | Abbreviation | Dynasty Name | Abbreviation |
| --- | --- | --- | --- | --- | --- |
| Stone Age | SQ | The period of the Sixteen States | SLG | Jin dynasty | Jin |
| Shang dynasty | Shang | Sui dynasty | Sui | Yuan dynasty | Yuan |
| Zhou dynasty | Zhou | Tang dynasty | Tang | Ming dynasty | Ming |
| Han dynasty | Han | The period of the Five Dynasties and Ten Kingdoms | WD | Qing dynasty | Qing |
| The period of the Three Kingdoms | SanG | Song dynasty | Song | Republic of China | MG |
| Jurchen Jin dynasty | JC | Liao dynasty | Liao | The People's Republic of China | GHG |
| The Northern and Southern dynasties | NBC | The Western Xia | XiX | | |

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
