# Peer review of "Analysis of Cultural Meme Characteristics for Big Data of Cultural Relics"

_information, doi:10.3390/info11120584_

Round 1
Reviewer 1 Report
The paper presents the cultural memes of dynasties extracted from the national collection of cultural relics data released by the State Administration of Cultural Heritage. The authors also used social network analysis to analyze the cultural characteristics of dynasties and elaborate their historical and cultural features.
The paper is nicely written, taking the reader through the problems. In what follows I provide some detailed comments on the improvements that in my opinion are necessary:
In Section 2 it is essential to provide some examples when introducing the notion of meme and their types
Please discuss in the conclusion section if there threats to validity of the results obtained from the proposed study.
In the introduction section I think authors should also take into consideration work exploiting ontologies when in line 49 consider “data gathered from text, images and sound”. For instance,
Briola et al., “Agent‐oriented and ontology‐driven digital libraries: the IndianaMAS experience”, Software: Practice and Experience 47 (11), 2017, 1773-1799
Additionally, the proposed study can be enriched by taking into consideration visual analytics approaches for cultural heritage, see the following survey:
- Windhager et al., "Visualization of Cultural Heritage Collection Data: State of the Art and Future Challenges," in IEEE Transactions on Visualization and Computer Graphics, vol. 25, no. 6, pp. 2311-2330, 1 June 2019, doi: 10.1109/TVCG.2018.2830759.
Other comments:
In the introduction section there are several problems with the references “Error! Reference source not found”
The reference numbers do not appear in the brackets []
Line 209: “(Figure 2 shows the peripheral area shows…” ???
“the from Sui dynasty to the Qing dynasty” -> “from the Sui dynasty to the Qing dynasty”
“In the Song dynasty, yellow was specially used by the royal family” this is not true according to Figure 2a. Probably is a different dynasty.
“arc, pattern” -> “arc pattern”
“dragon pattern, pattern, double dragon pattern” the name of the middle pattern is missing
It is not clear if the dynasties are listed in the figures in chronological order.
In Figure 8a it is difficult to read the names of the dynasties with dark backgrounds.
Author Response
We would like to express our sincere thanks to the reviewers for the constructive and positive comments.Please refer to the reply document for details.

Reviewer 2 Report
The paper address interesting topics, but there is lack of technical details describing the methods used for analysis of memes.
References contain number of publications, but looks more as list of publications, then real analysis of literature. Also numeric is not using standard forms.
The paper describe different aspect of memes, but methods of analysis, measurement of similarities, clustering methods are not described very well.
Also conclusion need to be update better and more clear present results, what are similarities among dynasties, etc.
Author Response

(The authors gave the same response as above.)

Round 2
Reviewer 2 Report
Dear authors thanks. Paper made big progress